# Spatial-Temporal Changes of Abarkuh Playa Landform from Sentinel-1 Time Series Data

Sayyed Mohammad Javad Mirzadeh [1,2] , Shuanggen Jin [1,3,*] and Meisam Amani [3,4]

1. Shanghai Astronomical Observatory, Chinese Academy of Sciences, Shanghai 200030, China; m.mirzadeh@shao.ac.cn
2. School of Astronomy and Space Science, University of Chinese Academy of Sciences, Beijing 100049, China
3. School of Surveying and Land Information Engineering, Henan Polytechnic University, Jiaozuo 454000, China; meisam.amani@woodplc.com
4. WSP Environment & Infrastructure Canada Limited, Ottawa, ON K2E 7L5, Canada
* Correspondence: sgjin@shao.ac.cn or sg.jin@yahoo.com

**Abstract:** Playas, as the flattest landforms in semiarid and arid regions, are extremely sensitive to climate changes, such as changes in the hydrologic regime of the landscape. The changes in these landforms due to irrigation, anthropogenic activities, and climate change could be a source of disasters. In this study, we explored the spatial-temporal changes of the Abarkuh Playa in Central Iran using the time series of the Sentinel-1 backscatter dataset in the three scales. Our results showed that the western area of the Abarkuh Playa has been changed to other landforms with different characteristics, which is clear from all backscatter maps. The spatial-temporal analysis of the time series of backscatter data using the independent component analysis and time series of precipitation revealed that the backscatter variations were associated with direct rainfall across the playa and the surface was reacting to changes in the soil moisture content. The results of the power scale showed that the boundary of the playa could successfully be recognized as the oscillating pattern from other landforms in the study area. Moreover, the spatial-temporal analysis of backscatter in the power scale showed that different polarizations could reveal different patterns of surface changes for the playa.

**Keywords:** Abarkuh Playa; radiometric terrain correction; spatial-temporal analysis; independent component analysis; SAR backscatter

## 1. Introduction

Playas are the flattest landforms, with less than 0.02% slope, which are mostly observed in interior desert basins and nearby coasts with semiarid and arid climates. Neal [1] has reported at least 50,000 playas in the world's semiarid and arid regions, with most extending less than three square kilometers. Playas are seasonally covered by water that gradually infiltrates the underlying aquifers or evaporates, resulting in salt and sediment deposition at the bottom and edges of the playas and forming their surface properties. For example, thick salts have created rugged crusts in the Devil's Golf Course in Death Valley, California, USA [2], and evaporative layers have resulted in a soft surface in the Bonneville Salt Flats in Utah, USA [3]. Deposited sediment is exposed to shrinking and drying, and the contained clay layers control sediment volume changes because clay-rich layers cause deep shrinkage and dryness of the sediment during long-term droughts. Playas are extremely sensitive to climate change and are influenced by changes in hydrologic regimes induced by climatic changes. However, changes in their area are also controlled by other factors, such as changes in groundwater inflow and evapotranspiration. Furthermore, playas are very vulnerable to wind and can be considered a source of dust hazard [4] (e.g., in Owens Lake, California, USA [5]; Urmia Lake, Iran [6,7]; and northwest China [8]). Since the hydrology of desert and semi-desert regions has been critically stressed due to the growing population and industrial and agricultural activities, investigating and mapping the spatial-temporal

changes of playas are necessary to assess how irrigation, anthropogenic activities, and climatic changes can influence the natural balance of these landforms [9] in desert and semi-desert regions and cause the disasters [4].

Generally, various landforms are mapped using field observation, pre-existing maps, and other collected data sources with different spatial and temporal resolutions, resulting in outdated, limited, and imprecise information [10–12]. The increasing number of remote sensing satellite systems with different spatial resolutions and repeat cycles, accompanied by the recent technological advancements in remote sensing methods and surveying techniques, have enabled us to access sufficient information on spatial distribution and temporal changes of landforms, surface and subsurface composition, and to prepare geomorphologic maps with a high level of accuracy at the small-to-large scales [13–16]. High-resolution satellite data are used for detailed geomorphological mapping, while low-resolution satellite images are used for global monitoring and reconnaissance mapping. Another advantage of satellite remote sensing is the deployment of hyperspectral sensors that can capture more comprehensive information across hundreds of bands (e.g., the Hyperion of the EO-1 satellite with 220 bands covering visible, short-wave infrared, and near-infrared). For example, hyperspectral data were well used for identifying minerals in regolith and surface deposits [17]. Additionally, remote sensing techniques provided insights into geomorphology through new applications [18,19], enhancement of measurement accuracy [20], new datasets to assess new ideas [21], development of data processing capability [22], imaging inaccessible regions [23], and advanced technologies such as airborne electromagnetics and radiometric data to offer information about landform composition and depth [24].

Active remote sensing systems (e.g., Synthetic Aperture Radar (SAR)) that can operate independently from daylight and weather conditions are great resources for mapping and monitoring landforms [25]. Compared to optical sensors, SAR sensors have several advantages, such as penetration of the signal into landforms to obtain subsurface information, sensitivity to moisture content and surface roughness to discover spatial-temporal changes in the landforms, and usage of cross-polarization data to attain detailed information about landform properties. Digital Elevation Models (DEMs), which are derived using the SAR dataset and the Interferometric Synthetic Aperture Radar (InSAR) technique, have widely been used for geomorphological mapping. Joint DEM-image data allow us to generate 3D image models and enable more powerful interpretation and visualization tools [26,27]. For example, Schneevoigt and Schrott [28] showed that alpine landforms could be detected using a multiscale and hierarchical classification derived from ASTER imagery and DEM. Smith and Clark [29] also reviewed different methods of DEM visualization for geomorphological mapping. DEM can also be used to calculate the geomorphometric parameters for quantitative analyses. For example, Burberry et al. [30] used stream networks derived from DEM data as a proxy index to define the underlying tectonic landform. Potts et al. [31] and Bubenzer and Bolten [32] also used the Shuttle Radar Topography Mission (SRTM) dataset to quantify dune morphologies within the Aeolian geomorphology. However, other products of SAR sensors, such as backscatter maps derived from the SAR data and coherence maps generated by the InSAR approaches, can be used to assess their potential in mapping the landforms and their spatial-temporal changes.

SAR backscatter has widely been used in various applications, such as crop monitoring [33–36], wetland mapping [37–39], land classification and change detection [40–42], disaster response and detection of natural hazards [43–48], and delineation of wet snow-covered areas [49]. Several studies also used the Sentinel-1 SAR data for the playa landforms to reveal morphodynamics and explore the evolution and surface properties. For example, Ullmann, et al. [50] used the Sentinel-1 data and InSAR coherence to determine the fluvial morphodynamics changes and to detect the surface disturbances and mass movements in the Damghan Playa, Iran. Moreover, Eibedingil et al. [51] used Sentinel-1 SAR images with InSAR techniques to identify land surface variations and sinks for sediment loading from the surrounding catchment in the Lordsburg Playa, New Mexico, USA. Given these recent studies, the present study is the first to investigate the spatial-temporal changes in

the Abarkuh Playa using Independent Component Analysis (ICA) and SAR backscatter time series data. Additionally, it combines the ICA results with the time series of climatic parameters to explore the driver and control of the spatial-temporal changes of natural balance in the Abarkuh Playa.

## 2. Materials and Methods

### 2.1. Study Area

The Abarkuh Playa is located in a district with a dry climate in central Iran between 30.82° and 31.24° N latitude and 50.41° and 53.84° E longitude. It covers an area of 1307 km$^2$, with an average elevation of 1459 m above mean sea level (m.s.l) and a slope of 2.8%, respectively (Figure 1). The playa is divided into five main parts: seasonal river, alluvial fans, dry mud flat, salty mud flat, and salty flat, and is mostly covered by evaporite sediments, i.e., chalk and salt formation, accompanied by saline clay. Figure 1 shows that the seasonal rivers and streams charge the playa from the eastern, western, and southern borders. Moreover, the geochemistry and mineralogy of the playa have strongly been influenced by the surrounding rock outcrops and rivers, which transport salts and sediments to the playa.

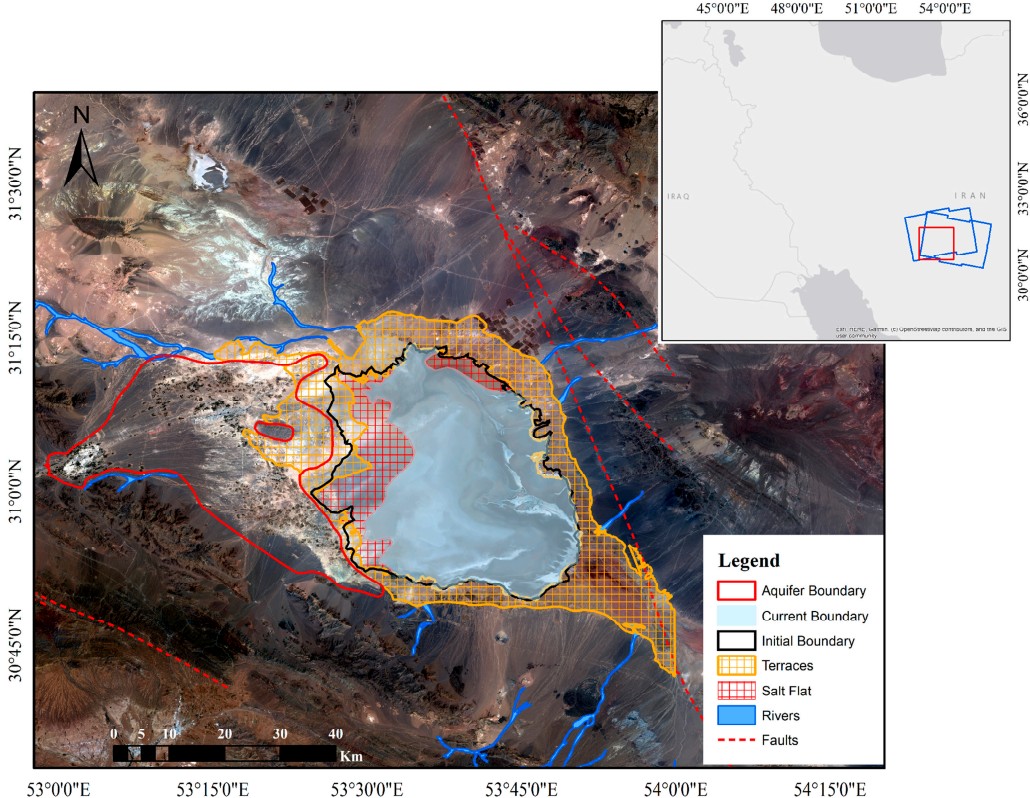

**Figure 1.** The geographical location of the Abarkuh Playa. Background: Sentinel-2 30 m RGB image. The red line shows the boundary of unconfined aquifer. The black line and blue-colored polygon demonstrate the initial and current boundaries of the playa, respectively. The initial and current boundaries of the playa are derived from the geological map at a scale of 1:250,000 in 2005 and up-to-date high-resolution optical imagery in 2022, respectively. The red dashed and blue lines indicate the faults and main rivers, respectively. The orange- and red-hatched polygons demonstrate the traces (alluvium) and Salt Flat, respectively. The inset map indicates the location of the Abarkuh Playa in Central Iran, shown by a red rectangle. The blue rectangles in the inset indicate the outline of frames from the ascending and descending dataset of Sentinel-1.

### 2.2. Datasets

2.2.1. Satellite Data

In this study, we used the Sentinel-1 Ground Range Detected (GRD) and Single Look Complex (SLC) dual-polarization (VV + VH) images from both ascending and descending orbits in the Interferometric Wide-swath (IW) mode. Sentinel-1 images were acquired between March 2017 and December 2022 from both descending and ascending orbits at a spatial resolution of 20 m by 5 m in the azimuth and range directions, respectively (outline of frames shown in the inset in Figure 1).

Sentinel-2A/B Multi-Spectral Imager (MSI) Level-1C (L1C) datasets that provide orthorectified Top-Of-Atmosphere (TOA) reflectance were also used to calculate the Sentinel-2 Water Index (SWI = (VNIR − SWIR)/(VNIR + SWIR)), where VNIR and SWIR correspond to Bands 5 and 11 of the Sentinel-2 image, respectively [52,53].

2.2.2. Geological and Meteorological Data

A geological map at a scale of 1:250,000, provided and distributed by the Geological Survey and Mineral Exploration of Iran (GSI) [54], was used to force the initial boundary of the Abarkuh Playa (see Figure 1).

The monthly average precipitation derived from the ECMWF Reanalysis v5 (ERA5)-Land hourly data was used to restrain weather data over the last few decades at a resolution of $0.1^{°} \times 0.1^{°}$ [55]. We used MOD11A1.006 Terra Land Surface Temperature (LST) and Emissivity Daily Global 1km products and Climate Hazards Group InfraRed Precipitation with Station Data (CHIRPS Pentad) [56,57], accessible through Google Earth Engine Data Catalog, to generate the long-term time series of LST (January 2001–January 2023) and Precipitation (January 1981–January 2023), respectively. CHIRPS is a 30+ year quasi-global rainfall dataset and incorporates 0.05-degree resolution satellite imagery with the in situ dataset to create gridded rainfall time series for trend analysis and seasonal drought monitoring [56]. MOD11A1.0006 product provides daily LST and emissivity values in a $1200 \times 1200$ km grid so that the temperature can be derived from the MOD11_L2 swath product [57]. The details of these two products are provided in Table 1.

**Table 1.** Information about the meteorological parameters.

| Name | Resolution | Units | Description |
| --- | --- | --- | --- |
| precipitation | 0.05 degree | mm | Precipitation |
| LST_Day_1 km | 1000 m | Kelvin | Daytime Land Surface Temperature |

### 2.3. Methods

2.3.1. Radiometric Terrain Correction

SAR data suffers from inherent radiometric and geometric distortions due to the side-looking acquisition in which the geometric distortions lead to geolocation errors in terrain features, and the radiometric distortions increase the uncertainties in any further analysis and applications. The main distortions of SAR data are described as follows [58]:

1. Shadow: When the back slope's angle is such that the sensor cannot image it entirely, it receives no information for a steep back slope;
2. Foreshortening: In this case, the backscatter from the front side of the mountain will be compressed altogether with returns from a large area arriving back to the sensor, which results in the front slope being shown as narrow;
3. Layover: In this case, returns from the back slope, the front slope, and part of the area before the slope arrived back to the sensor simultaneously. Thus, an area in the front of the slopes is projected onto the back side in the slant range direction of the image, and the data from the front slope is missed.

Radiometric Terrain Correction (RTC) addresses the geometric distortions of the SAR imagery using a DEM and adjusts the brightness or radiometry in the affected layover and

foreshortening regions through estimation of the actual area contributing returns to the sensor. Thus, RTC provides imagery with the values directly linked to the backscatter of the scene. During the RTC process, the input images are in the units of digital numbers (DNs), whereas the output images are in the units of $\beta_0$, $\gamma_0$, or $\sigma_0$, which are related to each other using the incidence angle ($\theta$) (Equation (1)).

$$\sigma_0 = \gamma_0 \times \cos\theta, \ \beta_0 = \frac{\gamma_0}{\tan\theta} \tag{1}$$

$\beta_0$ is the reflectivity per unit area in slant range that has not been corrected for incidence angle while $\sigma_0$ and $\gamma_0$ are the reflectivity per unit area, taking into account the incidence angle (from the metadata) and the local incidence angle through a DEM, respectively.

The Alaska Satellite Facility (ASF) has established Hybrid Pluggable Processing Pipeline (HyP3) service in which the pixel-area integration RTC approach is used [49] to process all Sentinel-1 SLC and GRD images to the fully geocoded and radiometrically terrain-corrected products [58,59]. The parameters of the RTC process of ASF's HyP3 service (see Table 2) can be adjusted for different applications. For example, the RTC products can be produced in three power, amplitude (square root of the power scale), and decibel (dB; 10 times the Log10 of the power scale) scales. Although the nature of three scales is the same, the power scale is appropriate for the statistical analysis of RTC dataset, which may not always be the best option for data visualization. In some cases, it may be desirable to convert the actual pixel values to a different scale [40,45].

**Table 2.** Customizable parameters of the RTC process in the ASF's HyP3 service.

| Parameters | Options | | |
|---|---|---|---|
| Radiometry | Gamma0 ($\gamma_0$) | | Sigma0 ($\sigma_0$) |
| Scale | Power | Decibel | Amplitude |
| Pixel Spacing | 30 m | | 10 m |
| DEM | Copernicus | | NED [1]/SRTM [2] |
| Co-registration | Dead Reckoning | | DEM Matching |
| Filtering | Do Not Apply | | Enhanced Lee Speckle Filter |

[1] National Elevation Dataset (NED), [2] Shuttle Radar Topography Mission (SRTM).

Using ASF's HyP3 service, we generated SLC and GRD RTC products at a pixel size of 30 m and in the unit of $\gamma_0$ and Universal Transverse Mercator (UTM) projection. We selected Copernicus DEM data for the RTC processing and geocoding. An enhanced Lee filter [60] was also applied to the datasets to decrease the SAR inherent speckle noise. The enhanced Lee filter uses the local statistics within individual filter windows to decrease speckle [61,62]. Finally, all three scales of products (power, amplitude, and decibel) were considered to assess the effect of different SAR scales in this study.

### 2.3.2. Independent Component Analysis

ICA has recently been used in studies to separate linear combinations of components, which are statistically independent and follow the non-Gaussian probability distribution from a mixed source, such as time series of deformation [63]. The connection between the original signal and independent components is described as (Equation (2)):

$$O_{n \times p} = D_{n \times l} \times S_{l \times p} \tag{2}$$

where $D$ is a mixing matrix in which each column associates to the coefficients of the contribution of each independent component; $S$ is the decomposed source matrix of the original observation matrix in which each row belongs to an independent component; $O$ is

the matrix of original signal; and *l*, *n*, and *p* are the number of independent components, the SAR epochs; and the pixels at each epoch, respectively.

In this study, we used the fast fixed-point algorithm, FastICA [64], to apply the ICA approach to the Sentinel-1 SLC and GRD data in the power, amplitude, and decibel scales, which have 974,492 samples per epoch, and 160 and 173 epochs (between 2017 and 2022) for the descending and ascending orbits, respectively. The FastICA algorithm whites and centers the original times series data by preconditioning them with Principal Component Analysis (PCA) and the truncation of variance rule to impose the number of independent components (ICs) and produce an orthogonal mixing matrix. Thus, the algorithm linearly transfers the original signal to be expressed as statistically independent components with a variance equal to 1 by whitening. The mix is represented by $W$, and the problem turned to $W = \hat{D} \cdot S$, where $\hat{D}$ is the orthogonally adjusted mixing matrix. Then, the source matrix is estimated by equation $S = \hat{D}^{-1} \cdot W$, where $\hat{D}^{-1}$ is called the unmixing matrix $U$. Then, a fixed-point iteration algorithm built into FastICA is used through maximizing spatial non-Gaussian sources to derive the source matrix ($S$), mixing matrix, and unmixing matrix ($U$). The ICA results are a temporal eigenvalue for each IC to determine its magnitude at each epoch and a score map, which has been scaled by the contribution of retained components to the mixed source and highlights the pixels with the observed eigenvalue [65].

## 3. Results and Analysis

### 3.1. Variations of Precipitation and LST

The long-term trends of precipitation and LST in the Abarkuh Playa are analyzed. Figure 2 shows the time series of monthly and yearly precipitation from January 1981 to January 2023, as well as monthly, maximum, minimum, and average annual LST from January 2001 to January 2023. The Abarkuh Playa experienced an average annual LST of 36.79 °C during the period of study (2017–2022), ranging from 14.97 °C (January 2021) to 53.90 °C (July 2019), with the lowest LST in January (average LST in January: 17.92 °C) and the highest in July (average LST in July: 52.34 °C). Figure 2 reveals that the Abarkuh Playa experienced an increase in the annual LST by 0.12 °C/year from 2001 to 2021, while an increase in the global surface temperature by 0.86 °C/year has been reported [66,67].

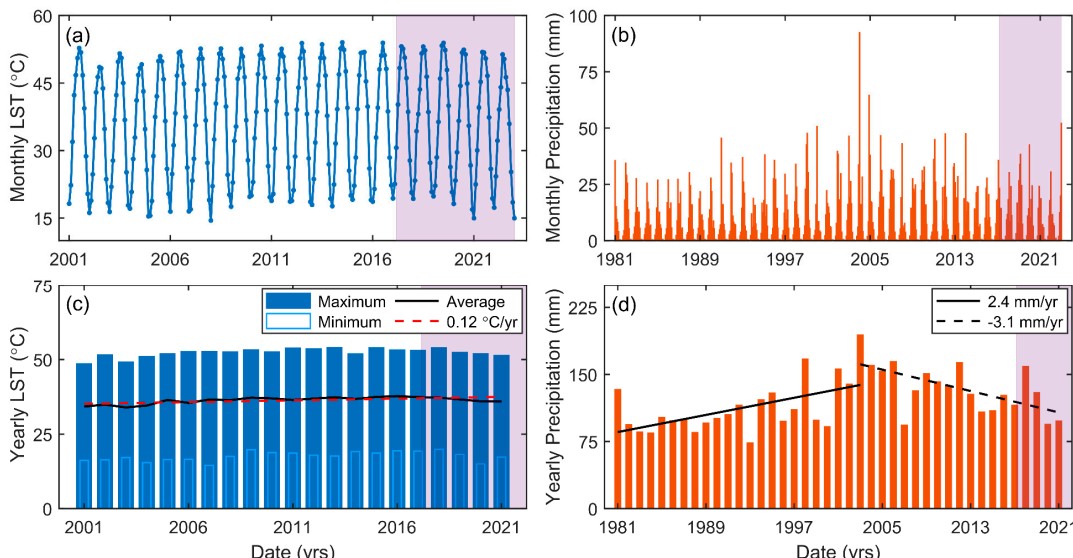

**Figure 2.** Time series of monthly (**a**) LST (January 2001–January 2023) and (**b**) precipitation (January 1981–January 2023). (**c**) Time series of maximum (blue-full bars), minimum (blue-edge bars), and average (black line) annual LST. The red dashed line shows the best fitting to the average annual LST with a slope of 0.12 °C/year. (**d**) Time series of annual precipitation. Black full and dashed lines show the best fitting to the time series between 1981 and 2003, and 2003 and 2021, respectively, with slopes of 2.4 mm/year and −3.1 mm/year. The purple-shaded time span in (**a**–**d**) shows the study period.

Figure 2d indicates that the annual precipitation ranged from 94 mm (2020) to 159 mm (2018) over a given hydrological year (defined as 1 October to 30 September) between 2018 and 2022. The majority of precipitation falls from October to May, while the minority of precipitation happens from June to September (see Figure 2b). Figure 2d shows that the playa experienced the maximum annual precipitation in 2003, while the precipitation had an increase and a decrease by 2.4 mm/year and 3.1 mm/year, respectively. It suggests that the Abarkuh Playa has experienced a critical situation in the recharge components for two recent decades (2003–2023), caused by the most severe historical droughts across the area [68].

### 3.2. Spatial Patterns of Backscatter

Figures 3 and 4 illustrate the spatial pattern of average SAR backscatter in three scales (i.e., amplitude, decibel, and power) derived from the 2017–2022 Sentinel-1 SLC and GRD dual-polarization ascending and descending dataset, respectively. The results reveal that the SAR backscatters from the Sentinel-1 SLC and GRD datasets have similar spatial patterns and values (see Figure 5a–l,m–x). It suggests that although SLC products are originally processed at a natural pixel spacing, and GRD products are multi-looked to reduce the impact of speckle noise, the type of Sentinel-1 products (i.e., SLC and GRD) does not influence the spatial analysis of the playa landform. Comparing the results of VV and VH polarizations shows that in all scales, VV polarization has more sensitivity to the landform and moisture changes than VH polarization inside the playa boundary, as well as in the northeast and southwest of the area covered by mountains and represents the higher values than VH polarization. Figure 3a,b,e,f,i,j displays that a part of the playa on the west side takes almost the same value of backscatter as the outside, suggesting that this part has changed to Salt Flat (shown by the red-hatched polygon in Figure 1) and possibly lost the playa characteristics. The low backscatter values are associated with old and high-to-medium levels of terraces (alluvium) across the plain, as demonstrated in Figure 3a,b,i,j, while the high values primarily belong to the playa and mountainous areas in the northeast and southwest of the region.

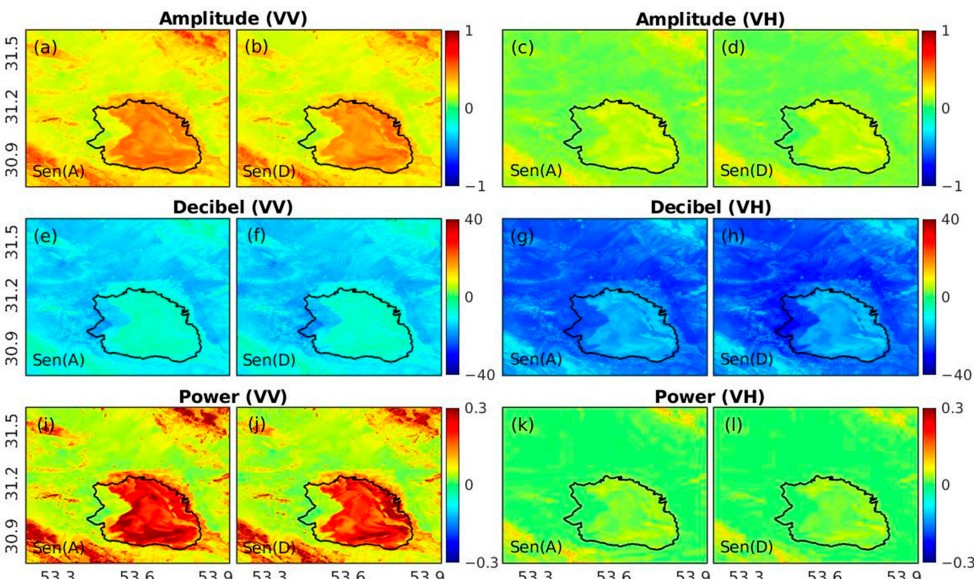

**Figure 3.** The maps of the average SAR backscatter in the (**a**–**d**) amplitude, (**e**–**h**) decibel, and (**i**–**l**) power scales, derived from the 2017–2022 Sentinel-1 SLC dual-polarization ascending and descending datasets. Black lines in (**a**–**l**) show the initial boundary of the playa. The Sen(A) and Sen(D) in (**a**–**l**) refer to the Sentinel-1 ascending and descending datasets, respectively.

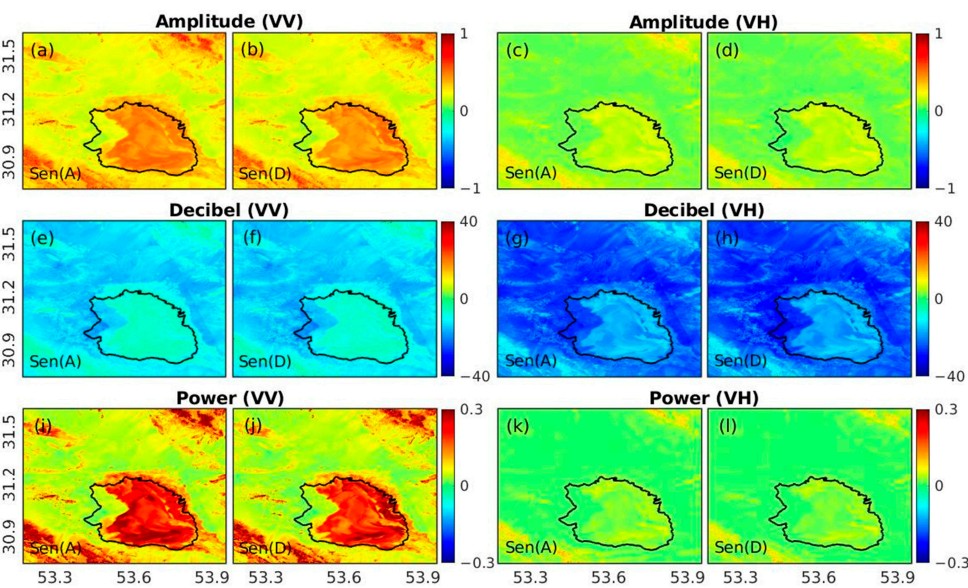

**Figure 4.** The maps of the average SAR backscatter in the (**a**–**d**) amplitude, (**e**–**h**) decibel, and (**i**–**l**) power scales, derived from the 2017–2022 Sentinel-1 GRD dual-polarization ascending and descending datasets. Black lines in (**a**–**l**) show the initial boundary of the playa. The Sen(A) and Sen(D) in (**a**–**l**) refer to the Sentinel-1 ascending and descending datasets, respectively.

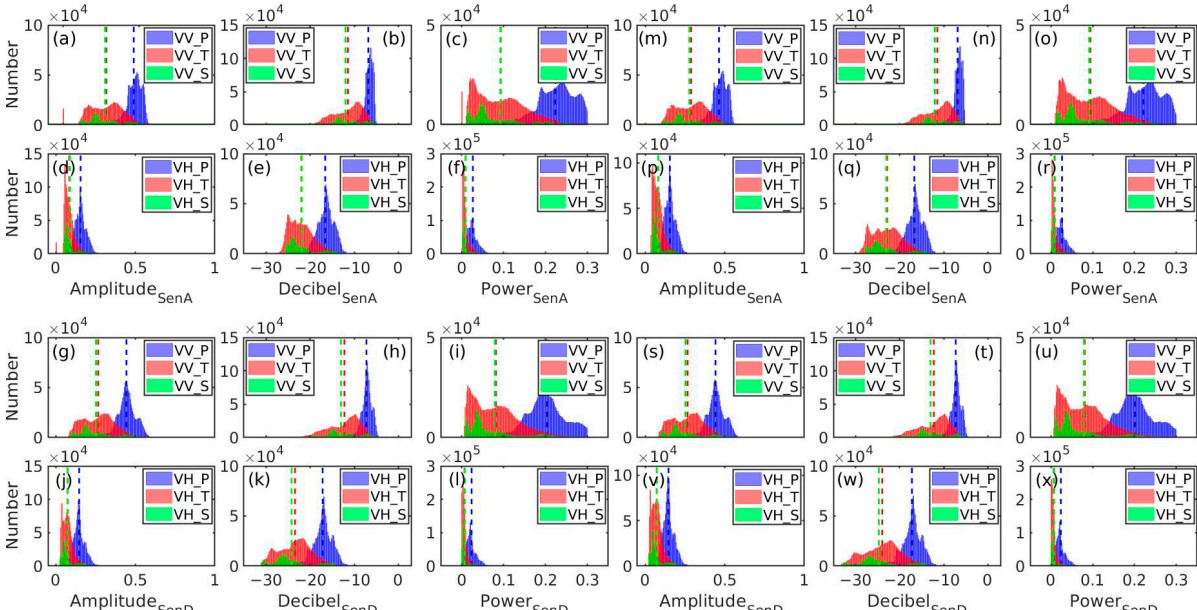

**Figure 5.** The histograms of the average SAR backscatter in the three scales of the amplitude (first and fourth columns), decibel (second and fifth columns), and power (third and sixth columns), derived from the 2017–2022 Sentinel-1 SLC (**a**–**l**) and GRD (**m**–**x**) dual-polarization ascending and descending datasets. Red, blue, and green colors indicate the backscatter values for the Terraces (T), Playa Lake (P), and Salt Flat (S) landforms, respectively. The Sen(A) and Sen(D) in (**a**–**x**) refer to the Sentinel-1 ascending and descending datasets, respectively.

We selected the backscatter values in a 40 m grid inside the boundaries of Terraces and Salt Flat (see Figure 1) and the current boundary of the playa for the three scales (i.e., amplitude, decibel, and power) and plotted their histograms in Figure 5. Figure 5 reveals that the average backscatter values for the Terraces and Salt Flat landforms are similar to each other in all cases, while the average backscatter values for the playa is different from two other landforms. In all three scales and landforms, the average and

ranges of backscatters from both ascending and descending are almost similar to each other, suggesting an insignificant effect of acquisition tracks on the ranges of backscatters. In the amplitude scale, the backscatters range from 0.45 to 0.55 for the playa and from 0.2 to 0.45 for the Terraces and Salt Flat landforms for the VV polarization. In the decibel scale, the backscatters range from −10 to −7 for the playa and from −20 to −10 for the Terraces and Salt Flat landforms for the VV polarization. In the power scale, the backscatters range from 0.1 to 0.3 for the playa and from 0 to 0.2 for the Terraces and Salt Flat landforms.

### 3.3. Seasonal Backscatter Changes

Figure 6 displays the time series of monthly precipitation generated from the spatial accumulation of the ERA5-land hourly data over the playa. It shows that the playa mostly experiences no rainfall from June to September each year (2017–2023) and has low-to-high rainfall rates in other months, well-correlated with the long-term time series of precipitation in Figure 2b. We considered June to September as the dry period and October to May as the wet period of the study area to explore how the backscatter changes spatially during the dry and wet periods. We divided the Sentinel-1 SLC VV polarization descending and ascending datasets into two groups associated with the wet and dry periods and calculated the temporal average backscatter across the study area, shown in Figure 7.

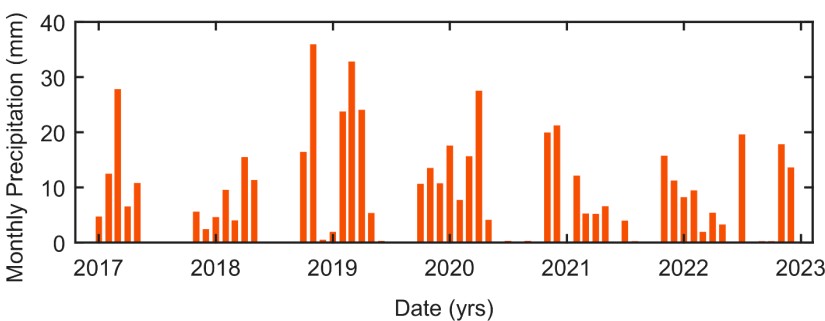

**Figure 6.** Time series of monthly precipitation, averaged spatially over the playa.

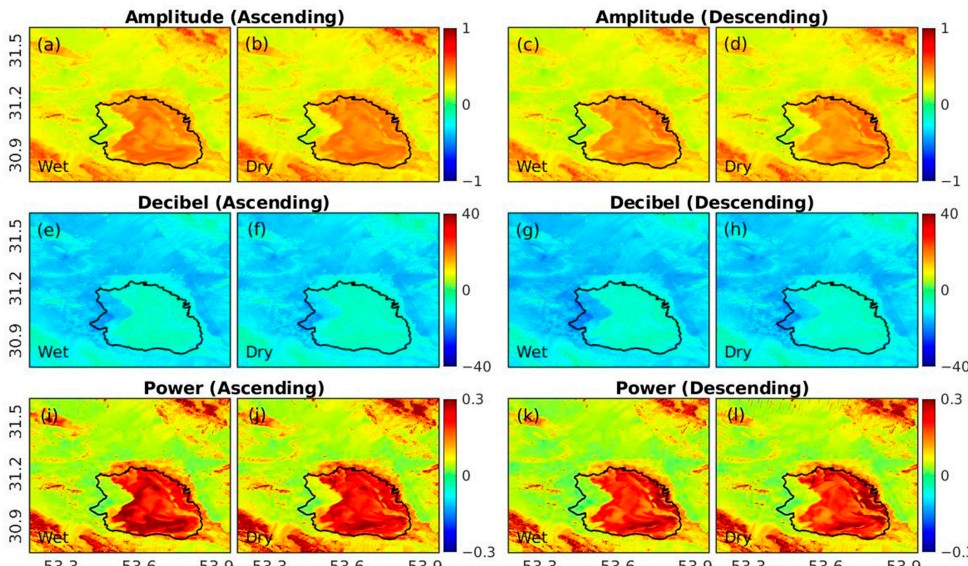

**Figure 7.** The maps of the average SAR backscatter in the (**a**–**d**) amplitude, (**e**–**h**) decibel, and (**i**–**l**) power scales, derived from the Sentinel-1 SLC VV polarization ascending and descending datasets. Black lines in (**a**–**l**) show the boundary of the Abarkuh Playa.

Based on the results illustrated in Figure 7, in both amplitude and decibel scales, there is no significant difference between the backscatter values during the wet and dry

periods inside the boundary of the playa. In the score maps of the power scale, a slight difference in the south to southwest of the playa is observed between the backscatter values during the wet and dry periods. This suggests that the direct rainfall over the playa cannot influence the significant changes in the surface properties, or the outflows through streams are late in reaching the playa, leading to changes in the backscatter values. Additionally, Figures 3 and 7 show that although the epochs of ascending and descending datasets are not the same, a significant difference cannot be seen in the backscatter values, suggesting that the average backscatter cannot possibly disclose slight changes in backscatter values caused by different polarizations (i.e., VV and VH) and acquisition tracks (i.e., ascending and descending).

*3.4. Controls on Spatial-Temporal Variations of Backscatter*

To explore the spatial-temporal variations of backscatter, we resampled the time series of SAR backscatter in three different scales (i.e., amplitude, decibel, power) into a 40 m grid, extracted 974,492 samples per epoch and 173 and 160 epochs for the ascending and descending datasets, respectively. Then, the datasets were imported into the ICA in the form of two-dimensional matrixes for each dataset with the size of the number of epochs $\times$ the number of samples. As discussed in Section 2.3.2, we applied the FastICA [64] to solve an ICA. The results for each IC include the temporal eigenvectors to show the signal magnitude at each epoch and the score map, scaled by the contribution of retained ICs to the original data, showing the pixels that are experiencing the observed temporal eigenvectors.

A single component explains more than 92% of the variance for all twelve resampled datasets (i.e., three scales for dual-polarization of both ascending and descending orbits). Figure 8 shows the ICA score maps and the temporal eigenvector results using a single component for the amplitude, decibel, and power scales of the backscatter. The score maps of all scales show similar backscatter patterns from the ascending and descending datasets inside the boundary of the playa, except in the amplitude scale of the VV polarization (Figure 8a,b). In all scenarios, the backscatter differences between the playa landform and other parts of the plain (Terraces and Salt Flat in Figure 1) are observed, especially in the power scale that is clearer with almost two different patterns from the VV and VH polarizations. Moreover, the western parts of the playa are different from the rest of its area, suggesting that these parts have been changed to other landforms with various natural and surface properties but exploring the reasons and time of this change is still challenging due to the short time of used data. In the VH polarization, both score maps and temporal eigenvectors from the amplitude, decibel, and power scales of the ascending and descending datasets are consistent with each other, while in the VV polarization, the score maps show a similar pattern with the differences in the temporal eigenvectors between the ascending and descending datasets. The mountainous parts in the northeast and southwest of the region show high score values, especially in the power scale in both VV and VH polarizations, which confirms the sensitivity of the backscatter to the moisture changes in high-altitude regions.

Although the temporal eigenvectors shown in Figure 8e,j,o seem to be noisy, the annual and seasonal patterns can still be seen in reaction to the changes in the parameters, which can change the SAR backscatter. Figure 9 shows the time series of precipitation over the playa with the temporal eigenvectors. It is obvious that by increasing the rate of precipitation during the wet period, the backscatter has increased, while in the dry period, the backscatter across the playa has quickly decreased due to no rainfall in the dry months. It confirms that the playa's surface and backscatter from the surface have reacted to (1) the direct rainfall over the playa area that was not obvious by focusing on the average SAR backscatter during the wet and dry periods and (2) outflows from the seasonal rivers and streams, which have arrived late to the Playa and recharged it. The delayed recharge and, as a result, changes in SAR backscatter can be seen in 2019 and 2020 in Figure 9c.

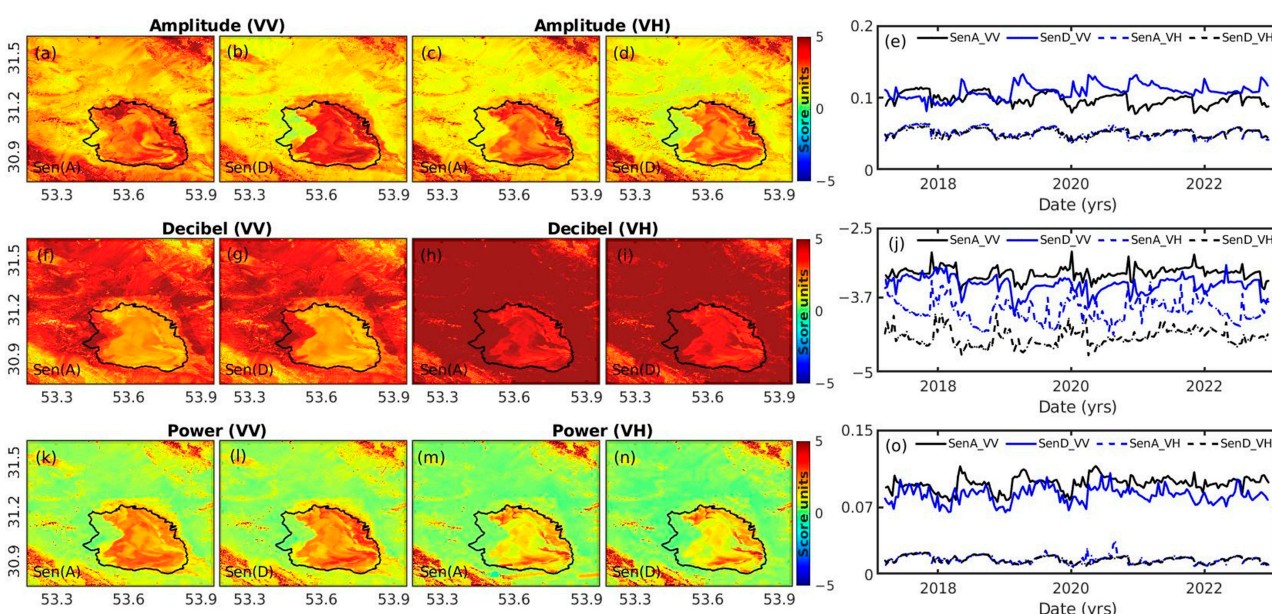

**Figure 8.** The ICA score maps and temporal eigenvectors result using one component for the (**a**–**e**) amplitude, (**f**–**j**) decibel, and (**k**–**o**) power scales of the Sentinel-1 SLC dual-polarization ascending and descending datasets. The black polygon in the score maps indicates the initial boundary of the playa.

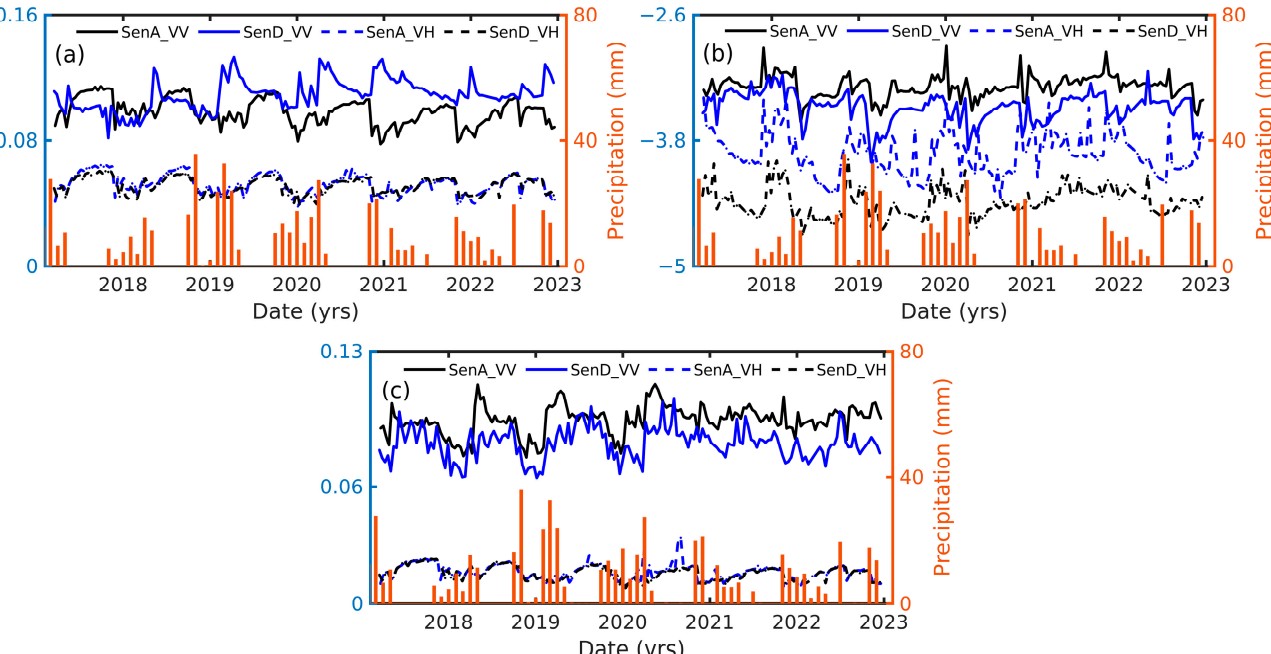

**Figure 9.** The ICA temporal eigenvectors result using one component for the (**a**) amplitude, (**b**) decibel, and (**c**) power scales of the Sentinel-1 SLC dual polarization ascending and descending datasets with the time series of monthly precipitation with the red-colored bars.

To investigate the changes in characteristics of the playa's surface revealed by the spatial-temporal analysis of backscatter using the ICA, we selected several epochs in Figure 9c, while the backscatter shows the peak and minimum values from 2018 to 2020. Table 3 shows the details of the selected epochs and the Sentinel-2 images with the minimum time interval with the selected epochs. Figure 10 displays the maps of the calculated SWI across the boundary of the playa in the selected Sentinel-2 epochs. Figure 10a–c reveals that during the dry epochs, the effects of moisture and inundation areas cannot be observed

inside the boundary of playa, while the SWI maps of wet epochs in Figure 10d–f show the inundation areas in the east and south of the area, affected by seasonal rivers and streams flows from the mountainous areas to the flattest part of the region.

**Table 3.** Details of the selected Sentinel-1 and 2 epochs to calculate the SWI maps shown in Figure 10.

| Dataset | Epochs | | | | | |
|---|---|---|---|---|---|---|
| | Dry Period | | | Wet Period | | |
| Sentinel-1 | 2018.03.03 | 2019.01.21 | 2019.11.29 | 2018.05.02 | 2019.04.03 | 2020.05.03 |
| Sentinel-2 | 2018.03.03 | 2019.01.20 | 2019.12.01 | 2018.05.02 | 2019.03.28 | 2020.05.04 |

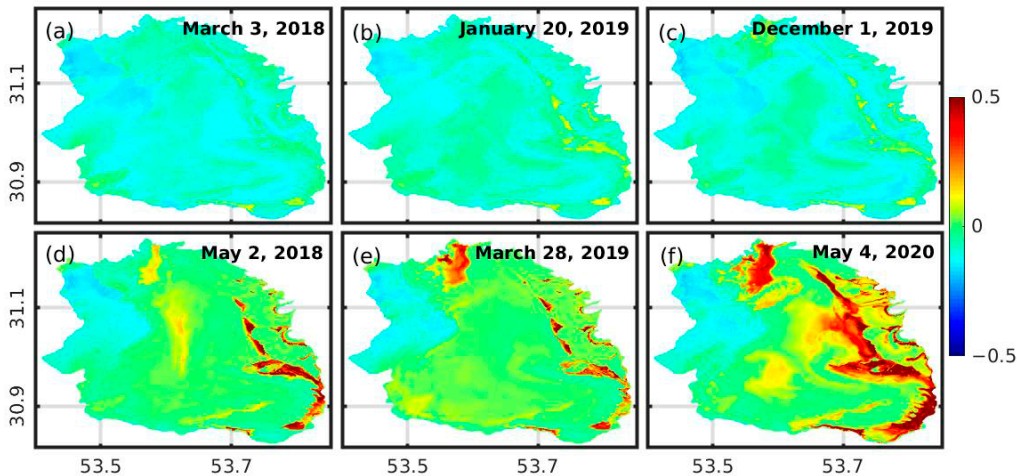

**Figure 10.** The maps of Sentinel-2 Water Index (SWI) for selected Sentinel-1 SLC epochs (see Table 3) in the (**a**–**c**) dry (no rainfall) and (**d**–**f**) wet (low-to-high rainfall) periods of the study area. Positive values show the water-effected or inundation areas inside the initial boundary of Abarkuh Playa.

## 4. Discussion

Playas are mostly formed from a mixture of salt, clay, and silt and seasonally covered by water that gradually infiltrates the underlying aquifers or evaporates, resulting in the salt and sediment deposition at the bottom and edges of the playas. Therefore, an oscillating reaction of rising and falling of the surface can be possibly observed due to the changes in the soil moisture contents. On the other hand, playas are the flattest landforms in the environment and are located in very low-altitude regions with less than a 0.02% slope. Therefore, their surface properties can be changed between the wet and dry seasons due to accumulate sediments transported by rivers and streams (Figure 1). The InSAR time series analysis could observe the oscillating pattern of rising and falling in the playa's surface, but there are uncertainties. For example, this oscillating pattern in the InSAR results may be interpreted as noise due to several sources of errors and uncertainties in the InSAR time series processing (i.e., unwrapping error and tropospheric effects). In the lack of time series dataset from controls and drivers, such as the volume of sediment transport by rivers, or low resolution of the parameters, such as soil moisture and precipitation, employment of other independent data can help to interpret the oscillating pattern of rising and falling in the playa's surface. The time series analysis of backscatters derived from the radiometrically terrain-corrected SLC or GRD datasets could be very helpful in interpreting the InSAR results and explore whether they represent the data or noise. Moreover, our results show that the results of spatial-temporal analysis of backscatter are completely compatible with the SWI calculated from the Sentinel-2 data in disclosure of the inundation areas as explored in the Black Rock Playa, Nevada, USA [69].

There is a common problem in SAR acquisitions called Radio Frequency Interference (RFI), which often occurs near airports or other infrastructures that emit strong radio/radar

signals. There are several studies with presenting methods for mitigation [70], but it is still challenging to solve the corresponding models, and the presented approach is mathematically complex. This problem will not cause serious issues or uncertainties in the InSAR processing and offset-tracking of SAR images, but it interferes with the analysis of the backscatter time series that works on the original SLC or GRD datasets. However, the analysis of the backscatter time series using the ICA displays that the ICA could successfully recognize and separate this issue from the affected epochs.

## 5. Conclusions

It is critical to investigate the geohazards and the surface characteristic changes triggered by climate changes in the desert and semi-desert areas and landforms, such as playas, using advanced solutions. In this study, we used the time series of SAR backscatter and precipitation across the Abarkuh Playa to discover the spatial-temporal changes of the playa's surface. The time series of backscatter in three scales (i.e., amplitude, decibel, and power) are analyzed from the Sentinel-1 SLC and GRD dual polarization ascending and descending datasets. The results show that the backscatters returned from the playa's surface are different from the other landforms (i.e., Salt Flat and Terraces) across the plain and have an oscillating pattern between the wet and dry periods. It was also observed that the temporal change in the backscatter time series in the boundary of the playa coincided with the changes in the time series of precipitation, suggesting that the playa's surface properties react to the direct rainfall, although it is charged by the inflows through the rivers and streams by a time lag. The results of the backscatter time series reveal that the oscillating pattern of rising and falling in the playa landform surface could be interpreted from the InSAR time series processing. Finally, the backscatter time series show a good consistency with the SWI maps calculated from the Sentinel-2 data, which show the inundation areas in the playa's surface between the dry and wet periods.

**Author Contributions:** Conceptualization, S.M.J.M.; methodology, S.M.J.M.; software, S.M.J.M.; validation, S.M.J.M.; formal analysis, S.M.J.M.; investigation, S.M.J.M.; resources, S.M.J.M.; data curation, S.M.J.M.; writing—original draft preparation, S.M.J.M.; writing—review and editing, S.J., S.M.J.M. and M.A.; visualization, S.M.J.M.; supervision, S.J.; project administration, S.J.; funding acquisition, S.J. All authors have read and agreed to the published version of the manuscript.

**Funding:** The Strategic Priority Research Program Project of the Chinese Academy of Sciences (Grant No. 600 XDA23040100) and the CAS-TWAS President's Fellowship supported this research.

**Data Availability Statement:** Publicly available datasets were analyzed in this study. The geological maps are accessible by contacting the Geological Survey and Mineral Explorations of Iran (GSI). The Sentinel-1 dataset is copyrighted by the European Space Agency (ESA) and freely accessible and pre-processed through the Alaska Satellite Facility (ASF) archive and ASF's Hybrid Pluggable Processing Pipeline (HyP3) service.

**Acknowledgments:** We thank the anonymous reviewers for their valuable comments on our manuscript. The authors thank the Geological Survey and Mineral Exploration of Iran (GSI) for the geological map of the study area.

**Conflicts of Interest:** The authors declare no conflict of interest.

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
