# Peer review of "Spatial-Temporal Changes of Abarkuh Playa Landform from Sentinel-1 Time Series Data"

_remotesensing, doi:10.3390/rs15112774_

Round 1

Reviewer 2 Report

Although your research topic is very interesting, I found several issues in the paper that need to be addressed before it can be published. Firstly, the English language used throughout the paper needs to be thoroughly revised. I recommend having a professional proofreader or reviser work on the manuscript to ensure it meets the required standard of scientific writing.

Secondly, The main problem is realated to the representation of the ground truth. A reable geomorphological and geological description of the area is missing. The playa needs to be better delineated to provide clarity and support of your research findings. We need a clearer description, map, or other details to aid the reader's understanding of the location's features. The authors acknoledge for a geologicla map but is not present in the paper.

Lastly, the relation with dust storm is unclear. We recommend providing more information or details to support your argument or hypothesis. Offering further explanations or modifications could strengthen your conclusions and results.

Reviewer 3 Report

See comments in attached file.  The conclusions need to be refocused to reiterate the novelty of the research, otherwise it is simply a case study.

Round 2

Reviewer 1 Report

To the editor,

Thanks to the authors’ efforts, all of my concerns have been addressed and I therefore recommend that the paper be accepted.